# Casein Kinase 1α as a Novel Factor Affects Thyrotropin Synthesis via PKC/ERK/CREB Signaling

**DOI:** 10.3390/ijms24087034

**Published:** 2023-04-11

**Authors:** Bingjie Wang, Jinglin Zhang, Di Zhang, Chenyang Lu, Hui Liu, Qiao Gao, Tongjuan Niu, Mengqing Yin, Sheng Cui

**Affiliations:** 1College of Veterinary Medicine, Yangzhou University, Yangzhou 225009, China; wangbingjie123188@163.com (B.W.);; 2Institute of Reproduction and Metabolism, Yangzhou University, Yangzhou 225009, China; 3Joint International Research Laboratory of Agriculture and Agri-Product Safety, The Ministry of Education of China, Yangzhou University, Yangzhou 225009, China; 4Jiangsu Co-Innovation Center for Prevention and Control of Important Animal Infectious Diseases and Zoonoses, Yangzhou 225009, China

**Keywords:** thyroid-stimulating hormone, levothyroxine, casein kinase, pituitary gland, signal transduction pathways

## Abstract

Casein kinase 1α (CK1α) is present in multiple cellular organelles and plays various roles in regulating neuroendocrine metabolism. Herein, we investigated the underlying function and mechanisms of CK1α-regulated thyrotropin (thyroid-stimulating hormone (TSH)) synthesis in a murine model. Immunohistochemistry and immunofluorescence staining were performed to detect CK1α expression in murine pituitary tissue and its localization to specific cell types. *Tshb* mRNA expression in anterior pituitary was detected using real-time and radioimmunoassay techniques after CK1α activity was promoted and inhibited in vivo and in vitro. Relationships among TRH/L-T4, CK1α, and TSH were analyzed with TRH and L-T4 treatment, as well as thyroidectomy, in vivo. In mice, CK1α was expressed at higher levels in the pituitary gland tissue than in the thyroid, adrenal gland, or liver. However, inhibiting endogenous CK1α activity in the anterior pituitary and primary pituitary cells significantly increased TSH expression and attenuated the inhibitory effect of L-T4 on TSH. In contrast, CK1α activation weakened TSH stimulation by thyrotropin-releasing hormone (TRH) by suppressing protein kinase C (PKC)/extracellular signal-regulated kinase (ERK)/cAMP response element binding (CREB) signaling. CK1α, as a negative regulator, mediates TRH and L-T4 upstream signaling by targeting PKC, thus affecting TSH expression and downregulating ERK1/2 phosphorylation and CREB transcriptional activity.

## 1. Introduction

Thyroid-stimulating hormone (TSH) is a glycoprotein hormone secreted by the pituitary gland, consisting of alpha and beta polypeptide chains encoded by *CGA* and *TSHB*, respectively [1,2]. TSH is essential to maintaining mammalian physiology and basic metabolism; fluctuations in this hormone can have considerable effects on daily activities [3]. Thyrotropin-releasing hormone (TRH) and estrogen promote TSH secretion, whereas adrenocortical hormone and somatostatin inhibit TSH secretion. Aberrant TSH secretion is associated with many diseases, including hyperthyroidism and hypothyroidism [4,5,6,7].

*TSHB* transcription is regulated by multiple factors, the activities of which are promoted by TRH. Protein kinase C (PKC) induces *TSHB* expression by activating protein AP-1 expression at the transcriptional level. Moreover, calcium-/calmodulin-dependent protein kinase induces gene transcription via the cAMP response element binding (CREB) signaling pathway. Mitogen-activated protein kinase (MAPK) activates the transcription factor ETS-1 and induces *TSHB* transcription [8].

The regulation of relatively constant pituitary hormone secretion involves long-loop feedback mechanisms. Thyroxine (T3/T4) is one of the main negative regulators of TSH and mainly acts by binding to nuclear receptors (TRs), which recognize specific T3/T4 responsive elements (TRE) in the T3/T4 target gene promoters to activate or inhibit transcription [9,10,11]. However, T3/T4 effects are not limited to nuclear receptors. For example, integrin αVβ3 has a binding site for T3/T4 analogs, as well as for other small molecules [12]. These ‘rapid non-genomic effects’ involve the activation of extranuclear cell surface receptors and their downstream cell signal transduction pathways [13,14,15,16]. Nevertheless, non-genomic pathways that mediate pituitary TSH synthesis and excretion have not yet been characterized.

Protein kinases and phosphatases have been extensively described in the regulation of circadian systems in eukaryotes [17]. Casein kinase 1 and casein kinase 2 (CK1/CK2) protein kinases are essential for the function of biorhythms in eukaryotes and prokaryotes [18]. Family members of CK1 exist in eukaryotes, including yeast and humans, with mammals possessing seven members [19]. In various biological processes, the areas covered by CK1 include Wnt signal transduction, cell membrane transport, and cytoskeleton maintenance [20,21,22,23]. Recent findings suggest that the Wnt pathway is an indispensable signal in the regulation of neuroendocrine metabolism in mice [24,25]. TSH, as one of the key hormones secreted by the pituitary gland, also has rhythmicity [26]. It is, therefore, of particular importance to characterize the role of CK1α in pituitary endocrine cells.

Thus, the purpose of this study was to objectively evaluate the role of CK1α in regulating TSH synthesis with the administration of TRH and L-T4 or thyroidectomy to mouse anterior pituitary in vivo and in vitro. This study also identified potential targets for the clinical prevention and treatment of central thyroid diseases caused by hormone disorders.

## 2. Results

### 2.1. Casein Kinase 1α (CK1α) Is Expressed in Murine Pituitary Thyrotropes

We used real-time polymerase chain reaction (RT-PCR) and Western blotting (WB) to detect the mRNA and protein levels of CK1α, respectively, in multiple healthy mouse tissues. Both gene and protein expression levels of CK1α were higher in the pituitary gland than in the thyroid, adrenal gland, or liver (Figure 1a,b). Cell types expressing CK1α in the pituitary gland were identified using dual immunofluorescence staining for CK1α with GH, FSH-β, TSH-β, ACTH, and PRL. The results show that over 90% of TSH-β-positive cells expressed CK1α, suggesting that CK1α may play important roles in TSH synthesis and secretion.

### 2.2. CK1α Inhibits Pituitary TSH Synthesis

Adult male mice were treated with 1 mg/mL CK1α inhibitor D4476 or 5 mg/mL CK1α agonist pyrvinium (q.d./p.m/iv). After 2 weeks, the degree to which the drug treatment affected key organs was determined primarily based on the organ ratio and organ coefficient (Appendix A). Subsequently, the levels of pituitary *Tshb* mRNA, p-CK1α/CK1α protein, and active-β-catenin/β-catenin protein were assayed. RT-PCR showed that D4476 treatment increased pituitary *Tshb* mRNA expression by 40.6% (Appendix A), while WB revealed a 31.6% decrease in p-CK1α/CK1α protein level. In contrast, pyrvinium treatment decreased *Tshb* mRNA expression by 23.5% and increased p-CK1α/CK1α level by 42.8% (*p* < 0.05); however, neither D4476 nor pyrvinium had significant effects on active-β-catenin nor β-catenin expression levels (Figure 2a–d). These data indicate that CK1α regulates TSH synthesis but not via Wnt/β-catenin signaling.

To verify the in vivo findings, primary pituitary cells were cultured in 24-well plates for 12 h and then treated with 5, 10, 20, or 40 μM D4476, or 0.8, 1.25, or 2.5 μM pyrvinium for 12 h before *Tshb* mRNA expression and serum TSH protein were measured. Treatment with 5 μM D4476 did not significantly impact *Tshb* mRNA expression nor serum TSH concentrations; however, both were considerably increased following treatment with 10, 20, or 40 μM D4476 (Figure 2e,f; *p* < 0.05). However, when the concentration was greater than 10 μM, the cells presented a degree of damage, and the apoptosis level was significantly increased (Appendix A). Conversely, *Tshb* mRNA expression significantly decreased after 1.25 and 2.5 μM pyrvinium treatments (Figure 2g; *p* < 0.05).

### 2.3. CK1α Regulates the PKC/ERK/CREB Signaling Pathway and Affects TSH Synthesis

The classical PKC/ERK/CREB signaling pathway is involved in the synthesis of pituitary TSH [26]. To evaluate whether the PKC/ERK/CREB signaling pathway mediates the regulation of TSH with CK1α, we analyzed the phosphorylation levels of PKC, ERK, and CREB in the mouse pituitary gland after treatment with 1 mg/mL D4476 or 5 mg/mL pyrvinium. D4476 increased the expression of p-PKC/PKC, p-ERK1/2/ERK1/2, and p-CREB/CREB 1.6-, 5.0-, and 1.3-fold, respectively, whereas pyrvinium decreased their expression 1.6-, 1.5-, and 1.2-fold, respectively (*p* < 0.05; Figure 3a,b).

These results were confirmed in cultured primary pituitary cells, which were pre-treated with 12 μM PKC inhibitor HA-100 for 1 h, followed by treatment with 10 μM D4476 for 6 h (Appendix A). *Tshb* mRNA was measured using RT-PCR. HA-100 treatment inhibited the enhancing effect of D4476 on *Tshb* mRNA expression (*p* < 0.05; Figure 3c).

### 2.4. TRH Reduces CK1α Activity

TRH is a key hormone secreted by the hypothalamus that regulates pituitary TSH synthesis and secretion. To explore whether TRH is an upstream regulator of CK1α, mice were treated with 10 mg/kg TRH. After 30 min, pituitary *Tshb* mRNA and CK1α protein levels were measured using RT-PCR and WB. Compared with that in the control group, *Tshb* mRNA expression in the TRH treatment group increased 7-fold (Appendix A), whereas *CK1α* mRNA expression was decreased 1.4-fold (*p* < 0.05; Figure 4a), and the CK1α protein level was decreased 1.5-fold (*p* < 0.05; Figure 4b).

### 2.5. CK1α Regulates the TRH-PKC-ERK-CREB Signaling Pathway and Affects TSH Synthesis

TRH regulates downstream molecules via myriad signaling pathways. For example, Sun and colleagues reported that the PKC/MAPK and calmodulin pathways represent specific transduction pathways involved in TRH signaling [27,28]. In the current study, the administration of TRH in vivo was found to activate the PKC/ERK/CREB signaling pathway (Figure 5a), but the classical Wnt/β-catenin signal pathway was not regulated by TRH (Appendix A). In in vitro experiments, we first screened the treatment time of TRH (Appendix A) and found that the expression of *Tshb* mRNA was significantly increased and that *CK1α* mRNA was significantly reduced after TRH drug treatment (*p* < 0.05; Figure 5b). Additionally, to explore whether CK1α participates in the regulation of the PKC/ERK/CREB signaling pathway by TRH, the CK1α agonist pyrvinium was administered. The results show that pyrvinium blocked the TRH-induced upregulation of the PKC/ERK/CREB signaling pathway (*p* < 0.05; Figure 5c,d), indicating that TRH mediates the regulation of TSH via the PKC signaling pathway with CK1α.

### 2.6. L-T4 Enhances the Activity of CK1α

Thyroxine (T4) is an important negative regulator of TSH expression [29]. Levothyroxine is a member of a class of drugs commonly used to inhibit the synthesis of pituitary TSH in vitro and in vivo [30]. Therefore, the impact of L-T4 administration on CK1α expression in vitro and in vivo was investigated, with a thyroidectomy group serving as an additional control (Appendix A). Compared with that in the control group, p-CK1α/CK1α level was enhanced 1.6-fold in the L-T4 treatment group and decreased 1.4-fold after thyroidectomy in mice (*p* < 0.05; Figure 6a,b).

Primary pituitary cells were treated with 10 µM L-T4 for 6 h, and there was no significant difference in *CK1α* mRNA expression, while p-CK1α/CK1α protein expression was upregulated approximately 2.5-fold (*p* < 0.05; Figure 6c,d). These results are consistent with the findings in vivo.

### 2.7. CK1α Inhibits the Effect of L-T4 Signaling on TSH Synthesis

To determine whether L-T4 influences TSH synthesis via the classical PKC/ERK/CREB signaling pathways or Wnt/β-catenin signaling pathway in the pituitary gland, the phosphorylation levels of the key proteins in the pathway were measured using Western blotting in control, L-T4 treatment, and thyroid removal groups. L-T4 significantly reduced the abundance of p-PKC/PKC, p-ERK1/2/ERK1/2, and p-CREB/CREB proteins (*p* < 0.05; Figure 7a). Opposite results were observed in the thyroidectomy group (*p* < 0.05; Figure 7b). However, there was no significant change in the Wnt/β-catenin signaling pathway (Appendix A).

These in vivo results were confirmed by blocking the activity of CK1α in vitro. More specifically, primary pituitary cells were pre-treated with D4476 and then treated with L-T4 for 30 min (Appendix A). The downregulation of the PKC/ERK/CREB signaling pathway by L-T4 was blocked by D4476 (Figure 7c,d), indicating that L-T4 inhibits TSH-β synthesis by promoting the post-translational modification of CK1α to inhibit the activation of the PKC/ERK/CREB signaling pathway.

### 2.8. L-T4-αVβ3 Activates the Post-Translational Modification of CK1α, Inhibiting PKC/ERK/CREB Signaling to Downregulate TSH Synthesis

The relevant literature indicates that thyroxine interacts with αVβ3 integrin on the cell membrane via a non-genome-dependent pathway to produce a cascade of reactions [31] (Appendix A). To determine whether L-T4 regulates CK1α through the αVβ3 integrin receptor, the αVβ3 inhibitor Tetrac was added to primary pituitary cells 1 h before L-T4 treatment, and *Tshb* mRNA and p-CK1α/CK1α protein abundance were measured. Tetrac blocked the inhibitory effect of L-T4 (10 μM) on *Tshb* mRNA expression (Figure 8a). Moreover, Tetrac blocked the enhancing effect of L-T4 on p-CK1α/CK1α (Figure 8b) and the expression of p-PKC/PKC, p-ERK/ERK, and p-CREB/CREB (Figure 8c). The addition of Tetrac also blocked the ability of L-T4 to reduce the p-PKC/PKC, p-ERK1/2/ERK1/2, and p-CREB/CREB ratios.

## 3. Discussion

In this study, we confirmed that CK1α mediates the regulation of TSH by TRH and L-T4 via the PKC/ERK/CREB signaling pathway, in which L-T4 activates αVβ3 integrin receptors with rapid non-genomic effects and mediates the negative feedback regulation of TSH with CK1α (Figure 9).

CK1α regulation is primarily involved in the transport, sub-cellular localization, activation/inactivation, and degradation of substrates [32]. CK1α can localize to the nucleus, where it regulates mRNA metabolism [33]. Furthermore, GLIPR1 mediates CK1α redistribution from the Golgi to the cytoplasm and increases CK1 protein levels, which are crucial to the phosphorylation and destruction of β-catenin [34]. In situ hybridization images included in the mouse brain atlas show that CK1δ expression is only detected in hippocampus-forming regions, while CK1ε expression is detected in all brain regions, such as the cortex, olfactory region, hippocampus, hypothalamus, and thalamus; however, previous studies have not focused on the role of these proteins in pituitary tissue [35]. Previous studies showed that the expression of CK1α in pituitary tissue was relatively high [22,36,37]. However, the sparse literature on the differential expression of CK1α in different pituitary cell types indicates that CK1α may be involved in the synthesis of pituitary TSH. Our findings demonstrate that CK1α is co-localized with TSH in anterior pituitary cells in mice and that 90% of CK1α is located in the cytoplasm.

By activating and inhibiting (using D4476 and pyrvinium) CK1α activity in vivo and in vitro, we found that the effects of D4476 and pyrvinium on TSHβ synthesis were reflected by changes in CK1α autophosphorylation levels. Although CK1 members are deemed rogue kinases, as their enzymatic activity is not typically regulated [38,39], they can be regulated by post-translational modifications, including autophosphorylation. Specific kinase inhibitors are a common and effective tool for studying the function of kinases. Notably, small-molecule inhibitors and promoters targeting CK1α are now widely available. However, D4476 is the most effective and universally accessible CK1 inhibitor, and it can effectively enter cells and ensure high intracellular activity [40]. In contrast, pyrvinium is an established inhibitor of the Wnt pathway, enhancing CK1α kinase activity and steadying Axin expression [41,42]. However, another study reported that pyrvinium does not activate CK1α but rather enhances GSK3 activity and downregulates the Akt signaling pathway [43]. Therefore, additional studies are required to understand the pathways involved in the effects of these molecules. 

Protein kinases represent a class of kinases that can induce substrate protein phosphorylation modifications, which often cause significant changes in the biological characteristics of substrate proteins and play an important role in the functional regulation of cells. Earlier studies provided evidence of the autophosphorylation inhibition of CK1δ and CK1ε carboxyterminal residues [44,45,46]. These CK1 isomers have longer carboxyl extensions than CK1α and CK1γ isomers. Although self-phosphorylation is also known to exist in CK1α, detailed studies of autophosphorylation sites have not been reported, and evidence is not available on the effects of such autophosphorylation on the catalytic activity of this isomer [47]. Therefore, kinase-inactivated mutant construction and associated biological activity analyses are the most common and effective methods used in kinase function research. For CK1α point mutations, a satisfactory kinase-inactivation mutant can be obtained at the ATP binding site [48], and mutations in the 136th positions D to N (dominant-negative; D136N) of CK1α reduce its kinase activity by approximately 90% [49]. According to the existing in vitro phosphorylation experiments [50], CK1α can be inactivated using point mutations (CK1α (K46A)), and wild-type CK1α plasmid protein has kinase activity that can catalyze the autophosphorylation of CK1α, while the CK1α (K46A) mutant protein does not present phosphorylation activity.

Further analysis revealed that CK1α inhibits TSH synthesis via the PKC/ERK/CREB signaling pathway. The intracellular mechanisms regulated by CK1α include substrate recognition specificity [51,52]. To date, more than 140 substrates of CK1 isomers have been reported in vitro and in vivo, including PKC, as well as transcription and splicing factors, such as CREB [53]. CREB phosphorylation caused by DNA damage is highly sensitive to the CK1α inhibitor D4476 [54]. Moreover, CREB regulates the expression of genes encoding TSH subunits in pituitary tissue [55]. TRH promotes the gene promoter of the TSH-β subunit, inducing CREB phosphorylation and recruiting CBP to the transcription complex in the process [56]. Specific signal transduction pathways involved in TSH signaling include inositol triphosphate, which musters intracellular calcium ions to cause TSH release; this pathway also involves the calmodulin and PKC-MAPK pathway [8]. This finding is consistent with our results, indicating that CK1 is essential for the transduction of TSH synthesis.

TRH and L-T4 are key regulators of TSH. We found that the expression of CK1α in the pituitary gland of mice stimulated using TRH in vivo and in vitro was negatively correlated with *Tshb* expression, while L-T4 inhibited the synthesis of TSH by promoting the post-translational modification of CK1α protein. Moreover, CK1α was found to impact TSH synthesis as a negative regulator.

Furthermore, we found that the long-loop feedback regulatory effect of L-T4 on TSH is mediated by the αVβ3 integrin receptor with CK1α. Accumulating evidence highlights the importance of complete nuclear TR protein and its main ligand, thyroid hormone, in healthy cells, referred to as the genomic function of thyroid hormone [57]. In addition, T4 has a defined non-genomic effect; the αVβ3 receptor was highly induced by thyroid hormone analogs; however, these have been primarily investigated within cancer cells [58,59,60]. In the current study, CK1α had a fixed position in the cytoplasm of pituitary tissue sub-cells. Moreover, L-T4 treatment induced an increase in αVβ3 integrin receptor expression. Hence, we primarily focused on the effect of T4 plasma membrane receptors on integrin αVβ3. Two thyroid hormone-binding sites of αVβ3 integrin were described in the thyroid hormone receptor activity model [61]. The first appears to bind to T3 to activate phosphatidyl 3-creatine, while the second binds to T4 and stimulates MAPK (ERK1/2) with phospholipase C and PKC [14,62]. Additionally, we observed that high T4 concentrations inhibit the expression of PKC/ERK with integrin receptors in normal pituitary cells.

Nevertheless, certain limitations were noted in the current study. The thyroidectomy model could not be assessed using imaging and was only indirectly examined using hormone levels. Further analyses are required to elucidate the regulation of CK1α protein isoforms by different neurotransmitters, which might provide valuable insights into CK1α function.

In summary, abnormal TSH secretion is induced by thyroid disease or by pituitary gland or hypothalamic disease, and TRH and L-T4 are clinically used as important regulators of TSH. Many genetic and lifestyle factors are associated with thyroid disease, and TSH, as one of the indicators of thyroid function, interacts with thyroid hormones. In this study, CK1α was identified as a potential target for the control of central TSH level in an established model of hyperthyroidism and hypothyroidism. The data presented here show that CK1α mediates the regulation of TSH by TRH and L-T4, and that L-T4 increases the phosphorylation of CK1α through αVβ3 integrin receptors, thereby inhibiting pituitary TSH synthesis via the PKC/ERK/CREB signaling pathway. The L-T4-αVβ3/CK1α-PKC-ERK-CREB regulatory mechanism also has a certain reference value in the treatment of thyroid diseases.

## 4. Materials and Methods

### 4.1. Animals

Male ICR mice, 6–8-week-old with a body mass of 25–33 g, were purchased from Yangzhou University Medical Center (Yangzhou, China). Mice were kept in a room with a natural light cycle and were provided free access to commercial mouse diets and water throughout the experiment. Our entire protocol, from design to implementation, is based on the methods proposed by the National Research Council guidelines for the nursing and purpose of animals used as subjects of experiments; the study was also approved by Animal Protection and Use Council of Yangzhou University (SYXK2017-0044).

### 4.2. In Vitro Culture

Freshly dissected mouse adenohypophyses were digested with 0.1% collagenase II (Sigma, St. Louis, MO, USA) and 0.25% pancreatic enzyme (Sigma). Anterior pituitary cells were harvested and resuspended in Dulbecco’s modified Eagle’s medium/F12 (GIBCO-Invitrogen, Grand Island, NY, USA), supplemented with 10% fetal bovine serum and antibiotics (100 U/mL penicillin G and 100 µg/mL streptomycin). In total, 10 × 10^5^ cells per well were cultivated in a 24-well plate for 12 h at 37 °C and 5% CO_2_. For each assay, a minimum of 3–4 solitary trials were conducted, and every trial contained 3–4 replicates per treatment condition.

The profound impact of CK1α on TSH synthesis and secretion from anterior pituitary cells was determined using a CK1α inhibitor, D4476 (0, 5, 10, 20, and 40 μM; MCE, Shanghai, China), CK1α agonist, pyrvinium (0, 0.8, 1.25, and 2.5 μM; MCE), or thyrointegrin receptor antagonist (Tetrac; MCE).

### 4.3. In Vivo Analysis

The concentrations of D4476 and pyrvinium were increased to 1 mg/mL and 5 mg/mL, respectively, in mice based on the half inhibition rate provided by MCE in vitro experiments.

Mice were intraperitoneally injected with L-T4 once per day between 4:00 p.m. and 8:00 p.m., and their body weight was measured every 3 days. The control group (three sets of biological replicates, *n* = 3 each) was injected with 0.9% normal saline at a dose of 40 μg/100 g, while the treatment group was injected with L-T4 solution at 30–50 μg/100 g.

Mice were subjected to thyroidectomy after the administration of anesthesia (pentobarbital sodium) [63]. Surgery was performed under sterile conditions. In brief, the skin was disinfected with 75% alcohol, and all mice were placed in the supine position; then, a rolled-up autoclaved gauze was used to elevate the mouse neck. The anterior portion of the neck was vertically cut by 2 cm. Each parotid gland was divided while it was gently grasped with atraumatic forceps. The muscle was longitudinally separated to expose the trachea and thyroid glands. The thyroid gland is organized under the thyroid cartilage in the neck, and the left and right lobes of the thyroid gland are located on either side of the trachea, where it appears as a brownish-red butterfly; the thyroid gland was separated from the trachea with ophthalmic tweezers. At the end of the procedure, the exposed tissues were irrigated with saline; the muscles, lymphoid tissues, and salivary glands were restored to their normal position; and the wounds were closed using 4-0 silk sutures.

Twenty-one days later, blood was collected from the mouse ophthalmic vein; mice were killed by neck dislocation after ether inhalation anesthesia; and pituitary tissue was excised and washed in a Petri dish containing normal saline. The pituitary gland was separated in half using autoclaved ophthalmic scissors along the center; one half was used for genetic analyses, and the other, for protein analyses. Short-term storage was carried out at −20 °C, and long-term storage was carried out at −80 °C.

### 4.4. RT-PCR

Total RNA in pituitary gland or cultivated cells was extracted using TRIzol reagent (Takara, Nanjing, China) according to the manufacturer’s instructions. cDNA was synthetized using a reverse transcription reagent (Vazyme, Nanjing, China). SYBR Green Master Mix (Vazyme) was used to determine the gene expression level in the ABI PRISM 7500 sequence detection system (Applied Biosystems, Foster City, CA, USA). The transcript expression levels were standardized to the endogenous expression of *Gapdh*. All primer sequences are shown in Appendix A.

### 4.5. Immunohistochemistry and Immunofluorescence

Adenohypophyses (anterior pituitary), including those within the sphenoid bone, were harvested and fixed in 4% paraformaldehyde, dehydrated in alcohol, cleared in xylene before paraffin embedding, sectioned at 5 μm, and finally dewaxed. Microwave or high-pressure antigen repair was performed in 0.01 mol/L citric acid buffer for 16 min; then, the tissue was cooled to room temperature and washed with PBS three times for 5 min each. To block non-specific antigen–antibody binding, sections were subsequently blocked with 5% normal donkey or goat serum (according to the attributes of the secondary antibodies) for 1 h at 25 °C and then incubated with antibodies overnight at 4 °C. After washing, the sections were incubated with fluorescent conjugated secondary antibodies at 25 °C for 1 h and then incubated with 4′,6′-diamino-2-phenylindole for 10 min. Finally, the sections were investigated under a fluorescence microscope camera system (IX71; Olympus, Tokyo, Japan). Antibodies used in the study are shown in Appendix A.

### 4.6. Radioimmunoassay

The concentrations of T4/T3 and TSH in mouse blood and cell culture fluid were measured using a radioimmunoassay kit (Beijing northern biology school, Beijing, China), following the manufacturer’s instructions. Briefly, the sample was attached with 125I-TSH, TSH, 125I-T4/125I-T3, and T4/T3 antiserum and counted using a gamma counter. The lowest detectable concentrations of TSH and T4/T3 were 0.15 µIU/mL and 0.5/20 ng/mL, respectively. The inter- and intra-assay coefficients of variation were <10 and <15%, respectively.

### 4.7. Western Blotting

Proteins were extracted from mouse pituitary and cultured primary pituitary cells using radioimmunoprecipitation assay lysis buffer (P0013B; Beyotime Biotechnology, Shanghai, China) and phenylmethylsulfonyl fluoride (PMSF-1:100; ST506-1; Beyotime). The protein concentrations were assessed using BCA Protein Assay Kit (CW0014S; CWbi, Taizhou, China) based on the manufacturer’s instructions. Protein lysates were separated using 13% SDS-PAGE and then transferred to polyvinylidene difluoride membranes (Bio-Rad Laboratories, Richmond, CA, USA). The membranes were sealed with 5% (*w*/*v*) skimmed dry milk for 1–2 h and then incubated with the antibodies at 4 °C overnight. Then, the membranes were washed three times in the three-buffer solution containing Tween20 and further incubated with HRP at 25 °C for 1–2 h. After rinsing, antibodies were detected using a highly sensitive ECL chemiluminescent detection kit (E412-01; Vazyme, Nanjing, China) at 2–8 °C. A gray value analysis was then performed using ImageJ. The antibodies used in the study are shown in Appendix A.

### 4.8. Statistical Analyses

All statistical analyses were performed using GraphPad Prism (version 9.0). The *t*-test was used for single comparisons, and one-way analysis of variance, for multiple comparisons. A *p*-value of <0.05 was considered statistically significant. The data are expressed as means ± standard error of at least three independent experiments.

## Figures and Tables

**Figure 1 ijms-24-07034-f001:**
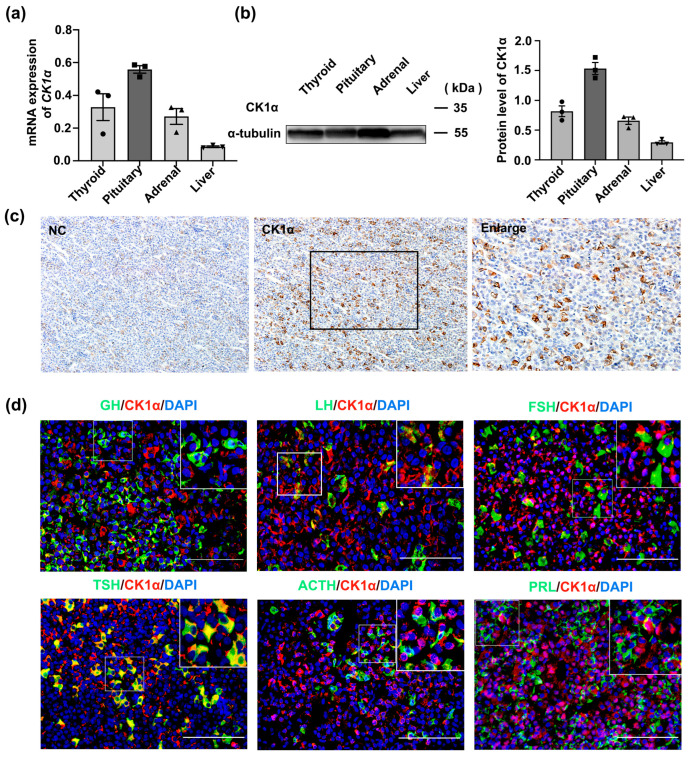
CK1α is expressed in murine pituitary thyrotropes. (**a**) The expression of CK1α in different mouse tissues was detected using real-time PCR (RT-PCR) and normalized to Gapdh expression. Data are presented as means ± standard error of the mean of three stand-alone experiments. (**b**) CK1α protein expression in different mouse tissues was analyzed using Western blotting and normalized to α-tubulin expression; *n* = 3. (**c**) The expression of CK1α in pituitary tissue was detected using immunohistochemistry. Scale bar = 100 μm. (**d**) Detection of CK1α expression in different pituitary cell types using immunofluorescence chemistry. Scale bar = 100 μm. Each experiment was independently repeated three times.

**Figure 2 ijms-24-07034-f002:**
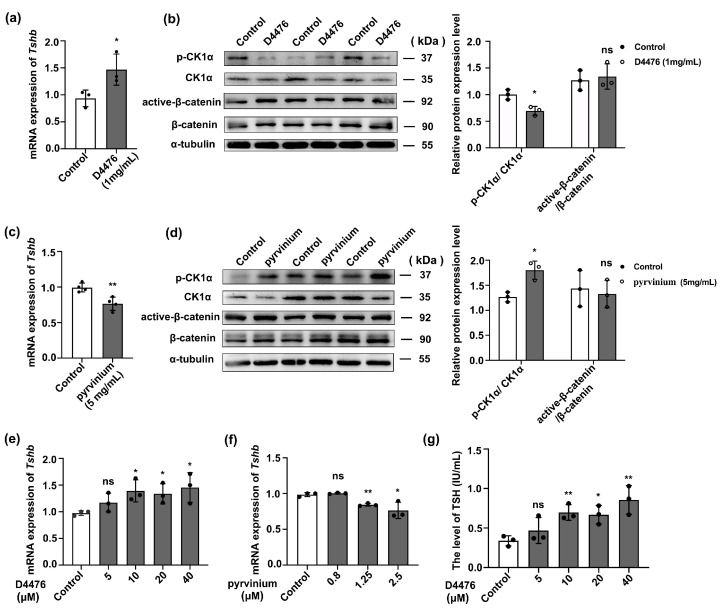
CK1α inhibits pituitary TSH synthesis. (**a**,**b**) The effect of D4476 on the expression of Tshb in mouse pituitary tissue was detected using the RT-PCR method, and the inhibition efficiency of p-CK1α/CK1α and active-β-catenin/β-catenin was detected using Western blotting. CK1α and β-catenin were used as the internal reference. * *p* < 0.05 and ns = no significant; *n* = 3. (**c**,**d**) After intravenous injection of pyrvinium, the effect of pyrvinium on the expression of Tshb in pituitary tissue was detected using RT-PCR. Western blotting was used to detect the induction efficiency of p-CK1/CK1α and active-β-catenin/β-catenin expression. CK1α and β-catenin were used as the internal reference. * *p* < 0.05, ** *p* < 0.01 and ns = no significant; *n* = 3. (**e**,**f**) The agonist and inhibitor of CK1α were added to primary cultured mouse adenohypophysis cells, and the expression of *Tshb* in vitro was measured using RT-PCR. *Gapdh* was used as the internal control. * *p* < 0.05, ** *p* < 0.01 and ns = no significant; *n* = 3. (**g**) The inhibitor of CK1α was added to primary cultured mouse adenohypophysis cells, and the secretion of TSH was measured using radioimmunoassay. * *p* < 0.05, ** *p* < 0.01 and ns = no significant; *n* = 3.

**Figure 3 ijms-24-07034-f003:**
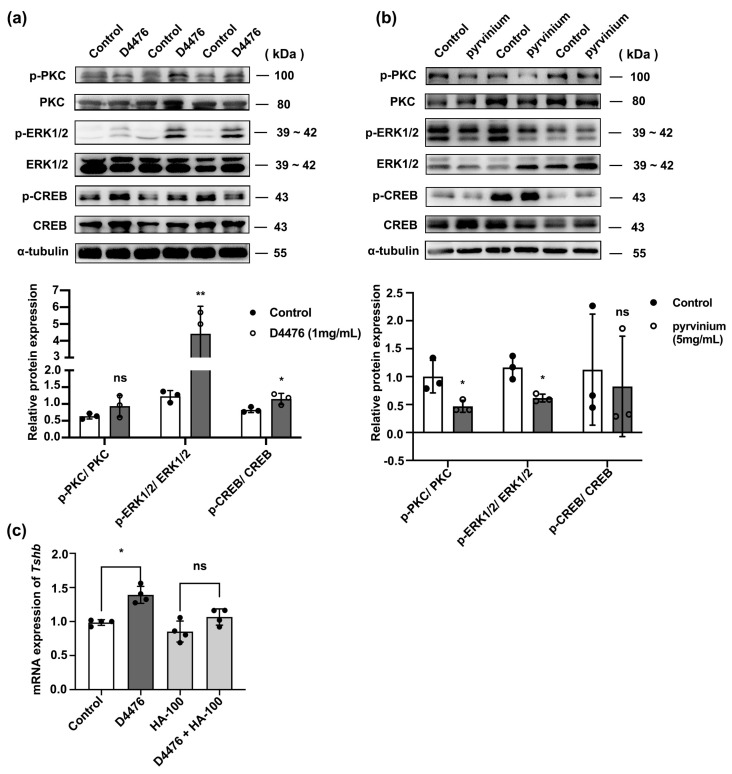
CK1α regulates the PKC/ERK/CREB signaling pathway upstream of TSH synthesis. (**a**,**b**) After intravenous injection of the CK1α inhibitor D4476 or activator pyrvinium, p-PKC, p-ERK1/2, p-CREB, PKC, ERK1/2, and CREB expression levels were analyzed using Western blotting. Relative p-PKC, p-ERK1/2, and p-CREB protein levels were normalized to those of total PKC, ERK1/2, and CREB; * *p* < 0.05, ** *p* < 0.01 and ns = no significant, n = 3. (**c**) The expression of *Tshb* was measured using RT-PCR in primary pituitary cells pre-treated with HA-100 and treated with D4476 for 6 h. *Gapdh* was used as the internal reference. * *p* < 0.05 and ns = no significant, n = 3.

**Figure 4 ijms-24-07034-f004:**
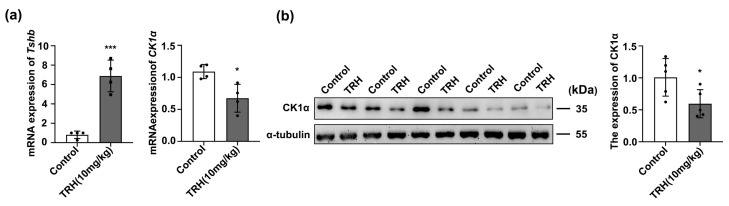
TRH reduces CK1α expression. (**a**) Pituitary tissue *Tshb* mRNA expression was quantified using RT-PCR after 1 h of TRH injection. *Gapdh* was used as an internal control. * *p* < 0.05 and *** *p* < 0.001; *n* = 3. (**b**) Western blotting was used to verify the expression of CK1α. The protein bands were normalized to that of α-tubulin. * *p* < 0.05, *n* = 5.

**Figure 5 ijms-24-07034-f005:**
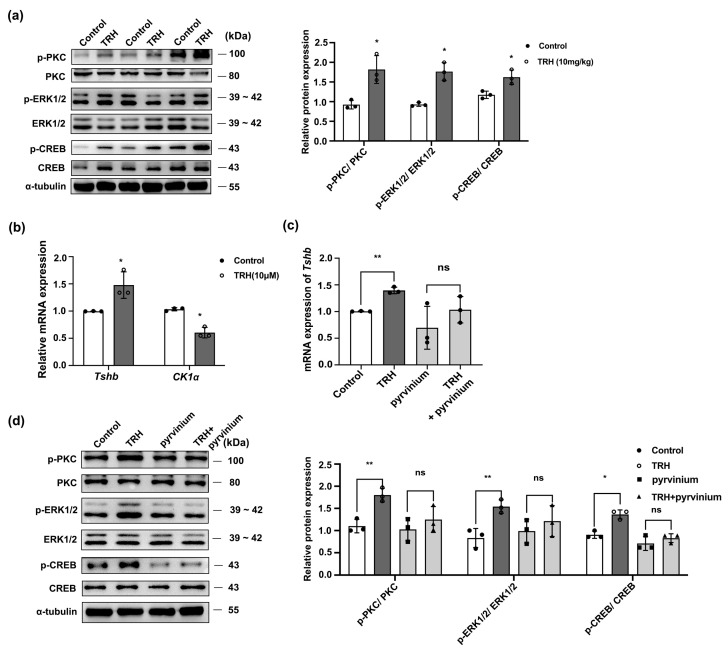
CK1α mediates the TRH-protein kinase C-extracellular signal-regulated kinase-cAMP response element binding (TRH-PKC-ERK-CREB) signaling pathway and TSH synthesis. (**a**) After injecting TRH (10 mg/kg) into mice, p-PKC, p-ERK1/2, p-CREB, PKC, ERK1/2, and CREB levels were assessed using Western blotting. Relative p-PKC, p-ERK1/2, and p-CREB protein levels were quantified after normalization to those of total PKC, ERK1/2, and CREB, respectively; * *p* < 0.05, *n* = 3. (**b**,**c**) *Tshb* mRNA and CK1α mRNA were detected by RT-PCR to determine the in vitro treatment time of TR. Primary pituitary cells were then pre-treated with pyrvinium for 1 h and then treated with 10 μM TRH for 30 min. *Tshb* mRNA was assayed using RT-PCR and normalized to Gapdh expression; * *p* < 0.05, ** *p* < 0.01 and ns = no significant, *n* = 3. (**d**) p-PKC, p-ERK1/2, p-CREB, PKC, ERK1/2, and CREB expression levels were determined using Western blotting. Expression levels were checked and measured by gray value assessment and normalized to total PKC, ERK1/2, and CREB as the internal reference; * *p* < 0.05, ** *p* < 0.01 and ns = no significant, *n* = 3.

**Figure 6 ijms-24-07034-f006:**
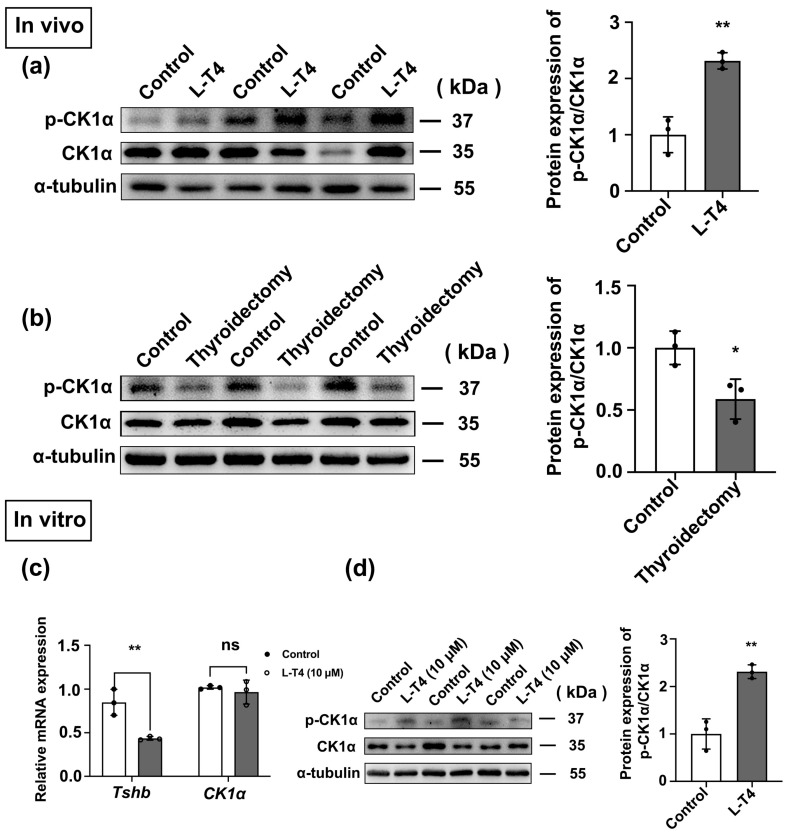
L-T4 enhances the activity of CK1α. (**a**,**b**) The expression of p-CK1α/CK1α in pituitary tissue was detected using Western blotting after the models of hyperthyroidism and thyroidectomy were established. The protein bands were normalized to CK1α. * *p* < 0.05 and ** *p* < 0.01; *n* = 3. (**c**) After primary pituitary cells were treated with L-T4 in vitro, *Tshb* mRNA expression was detected using RT-PCR. *Gapdh* was used as the internal reference. ** *p* < 0.01 and ns = no significant, *n* = 3. (**d**) In primary cultured mouse adenohypophysis cells, Western blotting was used to measure the effect of L-T4 on the expression of p-CK1α/CK1α. Western blotting bands were normalized to those of CK1α. ** *p* < 0.01; *n* = 3.

**Figure 7 ijms-24-07034-f007:**
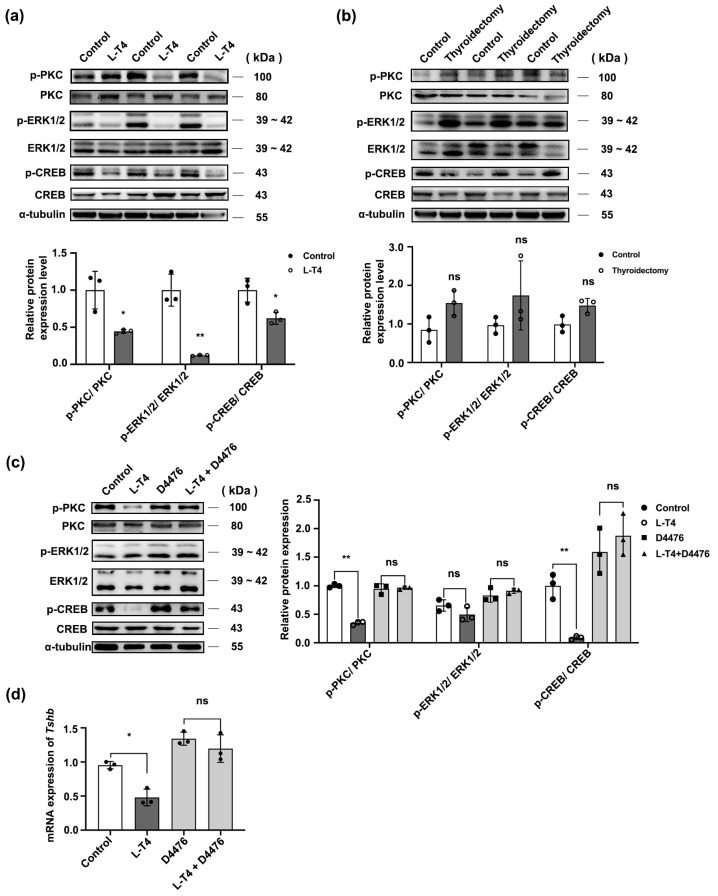
CK1α mediates the effect of the L-T4 signaling pathway on TSH synthesis. (**a**,**b**) In the L-T4 and thyroidectomy groups, p-PKC, p-ERK1/2, p-CREB, PKC, ERK1/2, and CREB abundance was determined using Western blotting; p-PKC, p-ERK1/2, and p-CREB were normalized to total PKC, ERK1/2, and CREB, respectively. * *p* < 0.05, ** *p* < 0.01 and ns = no significant, *n* = 3. (**c**,**d**) In vitro, pituitary cells were treated with D4476 and then with 10 μM L-T4 for 30 min; the *Tshb* transcriptional level was quantified using RT-PCR, and p-PKC, p-ERK1/2, p-CREB, ERK1/2, and CREB abundance was determined using Western blotting; PKC, ERK1/2, and CREB were used as the internal reference. * *p* < 0.05, ** *p* < 0.01 and ns = no significant, *n* = 3. Each experiment was independently repeated three times.

**Figure 8 ijms-24-07034-f008:**
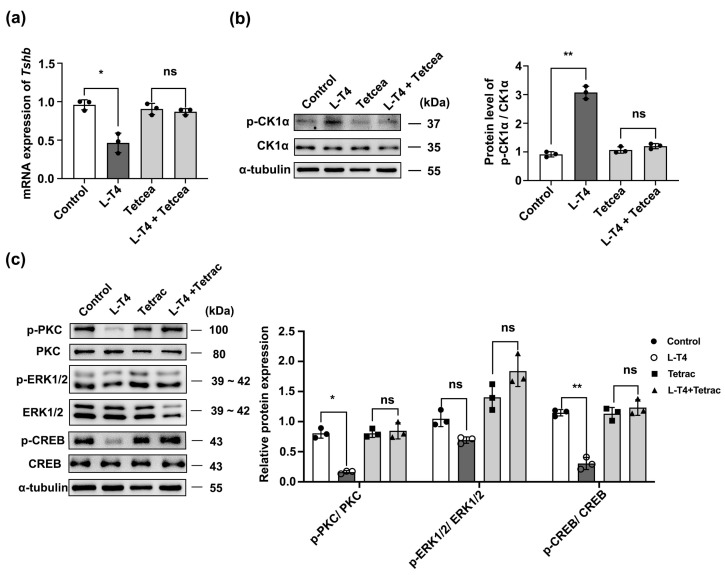
L-T4-αVβ3 activates the post-translational modification of CK1α to inhibit PKC/ERK/CREB signaling and downregulate TSH synthesis. (**a**) Relative *Tshb* mRNA levels after treatment for 1 h with 10 mM αVβ3 integrin inhibitor or αVβ3 integrin inhibitor pre-incubation followed by 10 mM L-T4 treatment. *Tshb* mRNA levels were normalized to *Gapdh*; * *p* < 0.05 and ns = no significant, *n* = 3. (**b**) Expression of p-CK1α/CK1α determined using Western blotting; CK1α was used as the internal reference. ** *p* < 0.01 and ns = no significant, *n* = 3. (**c**) Protein expression in primary cultured murine pituitary cells after pre-incubation with 10 mM αVβ3 integrin inhibitor or αVβ3 integrin inhibitor pre-incubation followed by 10 mM L-T4 treatment quantified using Western blotting and normalized to PKC, ERK1/2, and CREB expression levels; * *p* < 0.05, ** *p* < 0.01 and ns = no significant, *n* = 3. Each experiment was independently repeated three times.

**Figure 9 ijms-24-07034-f009:**
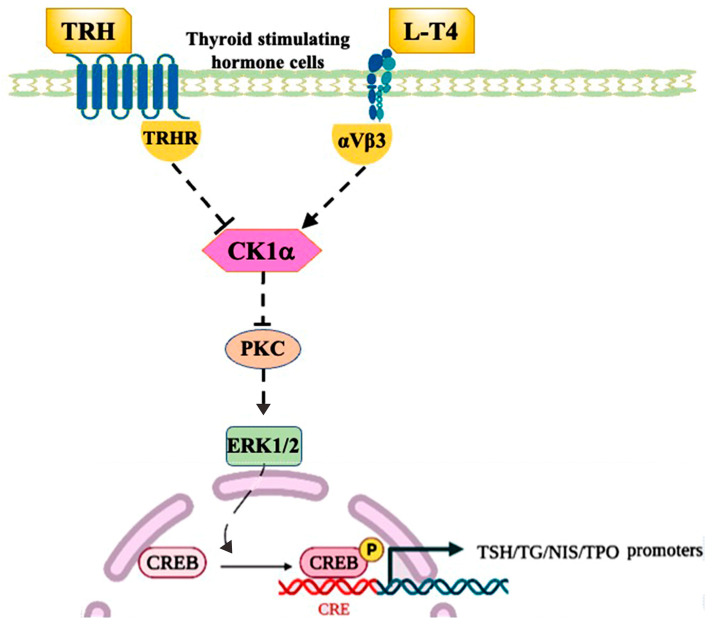
CK1α regulation of TSH. TRH activates PKC through the TRH receptor, which increases the phosphorylation of ERK1/2, thereby activating CREB and promoting the expression of genes encoding TSH subunits. L-T4 activates the integrin αVβ3 membrane receptor with a rapid non-genomic effect to inhibit PKC/ERK/CREB signaling, thus inhibiting TSH transcription. Arrows indicate stimulation; T bars indicate inhibition.

## Data Availability

The datasets generated during and/or analyzed during the current study are available from the corresponding author upon reasonable request.

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
