# Peer review of "Casein Kinase 1α as a Novel Factor Affects Thyrotropin Synthesis via PKC/ERK/CREB Signaling"

_ijms, 2023, doi:10.3390/ijms24087034_

Round 1
Reviewer 1 Report (Previous Reviewer 1)
The current study tried to figure out the regulatory mechanism of TSH expression in the mouse pituitary.
I don't think the authors have improved their manuscript.
As I pointed out previously, this manuscript has two critical issues;
1)lack of specificity in experimental design
The authors assume that the drugs (D4476 and pyrvinium) alter the PKC/ERK/CREB signaling through the CK1a activity. However, the drugs can affect the PKC/ERK/CREB pathway regardless of CK1a. They must rule out this possibility. Moreover, this study evaluated the CK1a activity with total pituitary. However, "non-thyrotrope" pituitary cells also express CK1a (Most pituitary cells are not thyrotropes). The exogenous stimuli (TRH or L-T4) can change the CK1a activity in "non-thyrotrope" pituitary cells. The authors must rule out this possibility.
2) large data fluctuation among replicate/triplicate samples
For example, in Figure 2D, the authors performed the experiment three times so that there are three controls. If they carried out the experiment in the same way, all those control bands should look the same. However, the control bands of p-CK1a are quite different from each other. Compared to the large difference among triplicate samples, the difference between the control and pp groups is minimal. If the authors claim that the experimental noise causes this, the noise is so huge that the data is not reliable.
Unless these fundamental problems are solved, it is impossible to draw a conclusion.
Author Response
Dear Review:
We thank you for giving us the opportunity to revise the manuscript, and we are extremely grateful to the editor and reviewer for the positive and constructive comments and suggestions on our manuscript. Please see the attachment.

Reviewer 2 Report (New Reviewer)
The main question is about the function and mechanisms of CK1 alpha regulated thyrotropin stimulating thyroid hormone, TSH synthesis in a murine model. The paper is well written, and the experiments are well conducted, however, some points must be discussed.
In terms of the drugs the authors assume changes in CK1 after treatment with D4476 and pyrvinium, however they don´t demonstrate if only these drugs could be affect the PKC/ERK/CREB signaling if not necessarily through CK1alpha.
The pituitary gland is divided into three sections-the anterior lobe which constitute most of the pituitary mass and is composed primarily of five hormone-producing cell types (thyrotropes, lactotropes, corticotropes, somatotropes and gonadotropes), I suggest some experiments to demonstrate that after exogen treatments the CK1 alpha activity not change in other cell types in the pituitary gland.
In my opinion the quality of figures could be improved. Many of them have poor quality. Figure 2 for example has some inconsistencies with respect to controls. In figures 4 and 5 maybe the authors can show more about the blot inside of the square. The blot is very close to the top of the square and the impression is that maybe more bands could be present in the experiments. In some figures the signal is too low, maybe the figure could be improved or the authors can use another technique to see the real effect. Finally, its hard to see the real effects in the experiments after treatments, in my opinion many of them need to be improved and also use other techniques to support the results.
Author Response
Dear Review:
We thank you for giving us the opportunity to revise the manuscript, and we are extremely grateful to the editor and reviewer for the positive and constructive comments and suggestions on our manuscript. Please see the attachment.

Reviewer 3 Report (New Reviewer)
This manuscript has the merit to characterize a novel mechanism controlling thyrotropin synthesis. In particular, the authors found that CK1alpha influences TSH expression via PKC/ERK/CREB axis. The experimental plan is well designed and the use of activators/inhibitors strongly support the causative link among the factors studied in this work. I only have few comments:
- Immunofluorescence in Figure 1D should be changed. Saturation is excessive and image aquisition appears artificial.
- What is PP in the graphs? I suppose pyrvinium. It should be clearly stated in the figure legends.
- Data are expressed as the mean ± S.E.M. The standard error of the mean indicates the uncertainty of how the sample mean represents the population mean. In my opinion, the authors inappropriately report the SEM instead of the Standard Deviation (SD). Since the SEM is always less than the SD, it deceives the reader into underestimating the variability between individuals within the study sample.
- I suggest to revise the working model (Figure 9) since the inhibitory arrows are not properly used. For instance, it seems that ERK activity inhibits CREB activation.
- The manuscript should be carefully revised by an english native speaker as several syntax and grammar errors are present throughout the text.
Author Response
Dear Review:
We thank you for giving us the opportunity to revise the manuscript, and we are extremely grateful to the editor and reviewer for the positive and constructive comments and suggestions on our manuscript. Please see the attachment.

Reviewer 4 Report (New Reviewer)
ID: ijms-2279573
Casein kinase 1α as a novel factor affects thyrotropin synthesis via PKC/ERK/CREB signaling. by Wang et al.
General comments:
In a murine model, the authors investigated the underlying function and mechanisms of CK1α-regulated thyrotropin (stimulating thyroid hormone, TSH) synthesis. They found that CK1α mediates the regulation of TSH by TRH and L-T4 via the PKC/ERK/CREB signaling pathway, in which L-T4 activates αVβ3 integrin receptors through rapid non-genomic effects and mediates negative feedback regulation of TSH through CK1α. It was considered that the topic was interesting, and the results included some novelty; however, some points should be addressed to improve the manuscript.
Specific comments:
1. The indicated symbols of PKC/ERK/CREB signaling in Fig. 9 seem incorrect. Arrows may be correct.
2. In the discussion, please add a description of the potential clinical applications and the future perspective of this study.
Author Response
Dear Review:
We thank you for giving us the opportunity to revise the manuscript, and we are extremely grateful to the editor and reviewer for the positive and constructive comments and suggestions on our manuscript. Please see the attachment.

Round 2
Reviewer 1 Report (Previous Reviewer 1)
The hypothalamus-pituitary-thyroid (HPT) axis is a critical endocrine system and is strictly regulated by multiple steps. The current study examines how the HPT axis is regulated at the pituitary level. The aim of this project is reasonable, given that the details of regulatory mechanisms in the HPT axis largely remain to be clarified.
The concern in this study is that there are critical issues pointed out in the previous comments. In the revised manuscript, the authors do not resolve these problems. Therefore, concluding that CK1a affects thyrotropin synthesis through PKC/ERK/CREB signaling is premature. Overall, this manuscript is preliminary.
This manuscript is a resubmission of an earlier submission. The following is a list of the peer review reports and author responses from that submission.
Round 1
Reviewer 1 Report
Hypothalamus and pituitary are central organs controlling the endocrine system. The pituitary secretes several essential hormones for growth, homeostasis, and reproductivity. An example is a thyroid hormone (TH) regulated by the hypothalamus-pituitary-thyroid gland (HPT) axis. A lot of studies reported the roles of thyroid hormone in peripheral tissues. However, the feedback mechanism in the HPT axis remains to be elucidated.
The authors in the current manuscript tried to figure out how TRH regulates TSH expression and how TH is involved in the negative feedback regulation in the HPT axis. They performed multiple experiments (both in vitro and in vivo) and concluded that CK1a regulates TSH synthesis through PKC/ERK/CREB signaling pathway, and TH mediates negative feedback regulation of TSH via CK1a.
The results they present might support the conclusion. But this study has critical issues that cannot be ignored.
- The experimental design lacks specificity. The authors need to rule out adverse effects caused by drugs (D4476 and pyrvinium). For example, these drugs could potentially affect the PKC/ERK/CREB signaling pathway independent of CK1a kinase activity, which could confound the results of the current study. More importantly, not only thyrotropes but other pituitary cell types express CK1a. Nonetheless, the authors evaluated the CK1a activity using the whole pituitary, which makes it difficult to conclude that CK1a specifically affects the PKC/ERK/CREB pathway to regulate TSH synthesis in the thyrotrope.
- In western blot analysis, the same amount of protein must be loaded in each lane. Otherwise, it isn't easy to interpret the result. For instance, the original images in Fig2B, Fig2D, Fig4B, Fig5A, Fig6A, Fig6D, and Fig7B show the different amounts of protein loaded in each lane. The authors claim the equal protein loading in each lane by showing the a-tubulin as a loading control, but there is an obvious difference even between controls of p-CK1a/CK1a or p-PKC/PKC or p-ERK/ERK or p-CREB/CREB. If the experimental noise causes this, the fluctuation of data is too huge, so that the results are unreliable. That is, the conclusion is not acceptable.
Reviewer 2 Report
The manuscript "Casein kinase 1α as a novel factor affects thyrotropin synthesis via PKC/ERK/CREB signaling" aims to shed light on the role of CK1α on hormonal metabolism. Nevertheless, it has some issues that should be addressed in order to improve its scientific quality:
The Y axis of the graph showing quantification of Western blot (Figure 1B) is entitled as "The expression of CK1α". Please, choose a more specific title showing that it is protein quantification. Same for other figures (for example Figure 8B).
In addition I would suggest to quantify immunohistochemistry in Figure 1C and comment on cellular types expressing each protein.
How was Western Blot quantification performed?
Regarding immunohistochemistry, more detailed information about the protocols is necessary in the Methods; antibodies, concentrations, etc.
Avoid the use of words such as neutral words such as "mediate" (Results subsection "2.7. CK1α mediates the effect of L-T4 signaling on TSH synthesis") and use more specific words such as "activates/inhibits" and in what conditions.
Reviewer 3 Report
This article by Wang et al. attempted to describe a mechanism that CK1a regulates thyrotropin synthesis. However, the conclusions are not supported by their data. There is a lack of basic understanding of CK1a function in cellular process. In addition, several experiments were poorly designed- comments see below.
1. I'm assuming the authors were using p-CK1a as an indicator of CK1a activity. What is p-CK1a? It's never introduced. However, authors have been using the relevant results as an important rationale of the work. If they meant phospho-CK1a, there is no current report suggesting p-CK1a in cells is linked to CK1a activity. Also, the antibody the authors used to detect p-CK1a is anti-AANAT, which has nothing to do with p-CK1a.
2. Line 63-64: CK1a in Neurospora is an ortholog of human/mouse CK1d, not CK1a. This dampens the base of this work- authors cannot use this as a part of the rationale why they started looking at CK1a.
3. All drugs in this work were used at concentrations that are several magnitudes higher than their EC/IC50s. In addition, even using at high concentrations, most effects the authors saw are subtle. Have authors ever determine if what they saw are off-target effects? For example, Figure 2E-G did not show dose dependent effects, which indicates off-target.
4. D4476 is a pan CK1 inhibitor, majorly used for CK1d/e inhibition. CK1d/e also plays a role in a lot of events, including Wnt signaling. However, authors have not taken this into consideration.
5. Authors performed intravenous injection for pyrvinium. However, multiple papers have suggested that pyrvinium has poor bioavailability so IV or IP injections won't be possible. Did authors determine if pyrvinium really went to the thyroid in their experiments?
6. Since authors were not using the proper drugs, I'm not sure if the effects on PKC/ERK/CREB are true. Even though it may be true, the PKC/ERK/CREB findings do not make sense. For example, in Figure 3 and probably Figure 5C, authors showed pyrvinium inhibits the phosphorylation of PKC, ERK, and CREB. However, this is not true in Figure 5D. Can authors explain why?
7. Some quantifications do not match the blots, e.g. Figure 5B, as most blots are overexposed and not really suitable for quantification. It's good for authors to show quantifiable results but this should match to original results.
8. Authors need to improve the logic flow when writing Results. Also, more details should be included in the figure legends for readers to understand.
Overall, I suggest to reject this paper. If the authors would like to resubmit, they have to fix the issues first.